

# Plant assemblages in atmospheric deposition

Ke Dong[1,2,3], Cheolwoon Woo[1], Naomichi Yamamoto[1,2]

[1]Department of Environmental Health Sciences, Graduate School of Public Health, Seoul National University, Seoul 08826, Republic of Korea
[2]Institute of Health and Environment, Seoul National University, Seoul 08826, Republic of Korea
[3]Major of Life Science, College of Natural Sciences, Kyonggi University, Suwon 16227, Republic of Korea

*Correspondence to*: Naomichi Yamamoto (nyamamoto@snu.ac.kr)

**Abstract.** Plants disperse spores, pollen, and fragments into the atmosphere. The emitted plant particles return to the pedosphere by sedimentation (dry deposition) and/or by precipitation (wet deposition) and constitute part of the global cycle
of substances. However, little is known regarding the taxonomic diversities and flux densities of plant particles deposited from the atmosphere. Here, plant assemblages were examined in atmospheric deposits collected in Seoul in South Korea. A custom-made automatic sampler was used to collect dry and wet deposition samples for which plant assemblages and quantities were determined using high-throughput sequencing and quantitative PCR with universal plant-specific primers targeting the internal transcribed spacer 2 (ITS2) region. Dry deposition was dominant for atmospheric deposition of plant
particles (87%). The remaining 13% was deposited by precipitation, i.e., wet deposition, via rainout (in-cloud scavenging) and/or washout (below-cloud scavenging). Plant assemblage structures did not differ significantly between dry and wet deposition, indicating that washout, which is likely taxon-independent, predominated rainout, which is likely taxon-dependent, for wet deposition of atmospheric plant particles. A small number of plant genera were detected only in wet deposition, indicating that they might be specifically involved in precipitation through acting as nucleation sites in the
atmosphere. Future interannual monitoring will control for the seasonality of atmospheric plant assemblages observed at our sampling site. Future global monitoring is also proposed to investigate geographical differences and investigate whether endemic species are involved in plant-mediated bioprecipitation in regional ecological systems.

## 1 Introduction

Approximately 374,000 plant species have been identified worldwide (Christenhusz and Byng, 2016), many of which release
spores, pollen, and plant fragments into the global atmosphere. An estimated 47–84 Tg of plant particles are released into the environment each year (Després et al., 2012;Hoose et al., 2010;Jacobson and Streets, 2009), where they have impacts at local and global levels. For example, inhalation of allergenic pollen can induce IgE-mediated hypersensitive reactions in sensitized individuals (D'Amato et al., 2007). Globally, atmospheric pollen influences climate by reflecting and absorbing



solar and terrestrial radiation (Spänkuch et al., 2000;Guyon et al., 2004), and/or by serving as ice nuclei (IN) and cloud condensation nuclei (CCN) (Pöschl et al., 2010;Pope, 2010). Finally, atmospheric pollen is involved in global cycling of substances (Després et al., 2012) by long-range transport and subsequent settlement to the planetary surface (pedosphere) by dry or wet deposition, i.e., sedimentation or precipitation, respectively.

5   Particle size influences the atmospheric processes experienced by airborne particles. Pollen grains are large in size (10–100 μm) compared with other biological particles such as viruses (0.02–0.3 μm), bacteria (0.3–10 μm), or fungal spores (0.5–30 μm) (Hinds, 1999;Jacobson and Morris, 1976). Pollen grains are thus more influenced by gravity than smaller particles and tend to settle rapidly (Aylor, 2002;Di-Giovanni et al., 1995). Large particles such as pollen grains serve as giant CCN (GCCN) that can efficiently collide and scavenge smaller droplets when settling from the atmosphere (Johnson, 10 1982;Möhler et al., 2007;Pope, 2010). Atmospheric pollen grains contribute to the formation and growth of cloud droplets and precipitation, and thereby influence the global hydrological cycle (Després et al., 2012;Pope, 2010).

   Dry and wet deposition processes are analyzed using different methods. Gravitational methods are used to quantify dry deposition of airborne pollen because of their rapid deposition velocities (Durham, 1946a;Yamamoto et al., 2015;Gong et al., 2017;Watanabe and Ohizumi, 2018), and volumetric methods are used to quantify amounts of pollen per unit volume 15 of air (Hirst, 1952;Núñez et al., 2017;Leontidou et al., 2018;Monroy-Colín et al., 2018). Precipitation (rain and snow) is analyzed for wet deposition of plant-associated materials such as phosphorus (Doskey and Ugoagwu, 1989) and organic carbon compounds from the atmosphere (Noll and Khalili, 1990;Mullaugh et al., 2014). However, pollen identification in the majority of aero-palynological studies is based on micromorphological characteristics (Levetin, 2004), which are subjective and limited in their resolving power. Recent studies have used DNA-based methods for accurate characterization of pollen 20 diversity and assemblages in the atmosphere (Leontidou et al., 2018;Núñez et al., 2017).

   Several studies have examined the emission of biological particles, including pollen, into the global atmosphere (Hoose et al., 2010;Jacobson and Streets, 2009;Elbert et al., 2007;Heald and Spracklen, 2009). Particle emission and deposition are balanced at the global level, and deposition can be used as a surrogate measure of particle emission, yet relatively few studies examining particle deposition have been conducted. Despite their quantitative advantages, DNA-based 25 techniques are not widely used for characterizing atmospheric pollen deposition, particularly where simultaneous sampling of wet and dry deposition is used.

   Our previous research showed that fungal assemblages differed significantly between wet and dry deposits, indicating taxon-specific involvement of fungi in precipitation (Woo et al., 2018), and we wished to determine whether plant assemblages displayed similar specificities. In this study, molecular techniques were used to compare the taxonomic 30 compositions and flux densities of plant particles in wet and dry atmospheric deposits. Additionally, we analyzed how allergy-related genera were deposited and removed from the atmosphere. This study gives insights into how plant



communities are involved in the cycling of substances through release of airborne particles such as pollen, spores, and fragments into the atmosphere.

## 2 Methods

### 2.1 Air and deposition sampling

Previously collected samples (Woo et al., 2018) were used for this study. Briefly, air and deposition samples were collected on the roof (approximately 20 m above ground level) of a building in a mountainous, forested area of Seoul in South Korea (37°27'55.0"N; 126°57'17.7"E). The sampling point was situated in a humid, continental, subtropical climate zone, according to Köppen climate classification. Approximately 4,000 species of spermatophytes (seed plants) inhabit South Korea (Korea National Arboretum, 2017), among which 1,048 and 1,500 species have been listed by the Korea National Arboretum (2017) and the Korea Research Institute of Bioscience and Biotechnology (2016), respectively. Samples were collected over 1 month periods during May to November 2015, with the exception of air sampling in August, which failed due to a severe rain event. A custom-made automatic dry and wet deposition sampler was used to collect atmospheric deposition, while an eight-stage Andersen sampler (AN-200; Sibata Scientific Technology Ltd, Tokyo, Japan) was co-located to volumetrically collect plant particles in the atmosphere. Substrates from the Andersen sampler were analyzed from three impactor stages corresponding to particles with aerodynamic diameters ($d_a$) of 4.7–7.0, 7.0–11, and >11 μm. The majority of plant DNA was detected from $d_a$ > 11 μm since pollen grains are large in size (10–100 μm). However, the stages for $d_a$ = 4.7–7.0 and 7.0–11 μm were also analyzed to detect remnant DNA that was not collected at the stage of $d_a$ > 11 μm due to inadequate sharpness of the particle collection efficiency curve of the impactor. Substrates loaded on the remaining stages were not analyzed due to difficulty in PCR amplification. Deposition sampling was conducted in duplicate, but PCR amplification was not possible in three of the seven dry deposition samples. In total, 18 air samples (6 months, 3 sizes), 11 dry deposition samples (7 months in duplicate, but 3 unsuccessful), and 14 wet deposition samples (7 months in duplicate) were amplified and subjected to DNA sequence analysis (Table S1 in the Supplement).

### 2.2 DNA sequencing

DNA extraction was performed as described previously (Woo et al., 2018). The internal transcribed spacer 2 (ITS2) region was amplified using universal plant-specific primers ITS-p3 and ITS-u4 (Cheng et al., 2016) with adapter sequences for Illumina MiSeq. PCR reaction mixtures (50 μL) contained 1 μL of extracted DNA, 0.4 μM each primer, 0.2 mM each dNTP, 1× reaction buffer, and 1.25 U Solg™ Taq DNA Polymerase (SolGent Co., Ltd., Seoul, Korea). Amplifications were performed using a T100™ thermal cycler (Bio-Rad Laboratories, Inc., Hercules, CA, USA) with the following thermal cycle: 10 min at 94°C (initial denaturation); 34 cycles of 30 s at 94°C, 40 s at 55°C, and 60 s at 72°C; and 10 min at 72°C



(final extension). The resultant amplicons were indexed using a Nextera XT Index kit (Illumina, Inc., San Diego, CA, USA) with the following thermal cycle: 3 min at 95°C (initial denaturation); 8 cycles of 30 s at 95°C, 30 s at 55°C, and 30 s at 72°C; and 5 min at 72°C (final extension). The indexed amplicons were purified using AMPure XP beads (Beckman Coulter, Inc., Brea, CA, USA), normalized to 4 nM with 10 mM Tris-HCl (pH 8.5), and pooled with 30% internal control

PhiX. Heat-denatured pooled amplicons were loaded onto a V3 600 Cycle-Kit reagent cartridge for 2×300 bp sequencing by Illumina MiSeq.

## 2.3 DNA sequence processing and analyses

Raw sequence reads were demultiplexed and trimmed for reads with a quality score <20 using MiSeq Reporter v2.5 (Illumina). Assembly, quality check, and taxonomic assignment of sequence reads was performed using USEARCH

v.11.0.667 (Edgar, 2010). Low-quality reads with >1.0 expected errors were removed, and joined reads of <200 bp were further excluded. Unique sequences were identified using default USEARCH settings. The UPARSE algorithm was used to remove chimeric reads, and the remaining reads were clustered into operational taxonomic units (OTUs) at 97% sequence similarity. In total, 1,261,572 reads from 43 libraries were mapped onto 97% OTUs (Table S1 in the Supplement). Taxonomic assignment was performed using the sintax algorithm with a cutoff value of 0.5 (Edgar, 2018) against the ITS2

database (Sickel et al., 2015;Ankenbrand et al., 2015). P-tests were performed using mothur v.1.39.5 (Schloss et al., 2009) to compare taxonomic structures. Reproducibility of assemblage structures was assessed based on biologically duplicated deposition measurements, with a statistical significance observed across the samples, but not within each sample (P-test, ParScore = 12, $p < 0.01$) (Fig. S1a in the Supplement). For α-diversity analyses, 6,142 reads were randomly sub-sampled from each library. The rarefaction curves appeared to reach asymptotes or near-asymptotes (Fig. S2 in the Supplement),

indicating that the sequencing depth was adequate for taxonomic richness estimation of the analyzed samples.

## 2.4 Quantitative PCR

Quantitative PCR (qPCR) was performed using universal plant primers ITS-p3 and ITS-u4 (Cheng et al., 2016) to quantify total copy numbers (CNs) of the ITS2 region. Reaction mixtures (20 μL) contained 1× Fast SYBR Green Master Mix reagent (Thermo Fisher Scientific, Waltham, MA, USA), 10 μM each primer, and 1 μL of extracted DNA. QPCR reactions were

conducted in triplicate using a QuantStudio™ 6 Flex Real-time PCR system (Applied Biosystems, Waltham, MA, USA) with the following thermal cycle: initial denaturation for 10 min at 95°C followed by 40 cycles of 15 s at 95°C and 60 s at 60°C. Calibration curves were generated using serial dilutions of a known concentration of PCR amplicons from a synthesized template containing an *Arabidopsis thaliana* ITS2 sequence (Unfried and Gruendler, 1990). The synthesized template was amplified with the primers ITS-p3 and ITS-u4 and quantitated using a DS-11 FX

spectrophotometer/fluorometer (DeNovix, Wilmington, DE, USA). Inhibition was considered as described previously





(Hospodsky et al., 2010), and no inhibition was observed. As previously described (Hospodsky et al., 2010), DNA extraction efficiency was estimated at 10% when reporting pollen quantities. QPCR measurements were confirmed to be biologically reproducible with a cumulative coefficient of variation of 62% on an arithmetic scale (Fig. S1b in the Supplement).

**2.5 Calculations**

Taxon-specific plant quantity was estimated by multiplying the DNA sequencing-derived relative abundance of each taxon by the total plant quantity by the universal plant-specific qPCR, as previously described (Yamamoto et al., 2014;Dannemiller et al., 2014;An et al., 2018). The calculated genus-level deposition flux densities were confirmed to be biologically reproducible with a cumulative coefficient of variation of 91% on an arithmetic scale (Fig. S1c in the Supplement). The annual dry deposition velocity ($V_d$) was estimated for each plant taxon according to the following equation:

$$V_d = \sum_{j=1}^{6} F_j \Big/ \sum_{j=1}^{6} \sum_{i=1}^{3} N_{j,i} \, , \tag{1}$$

where $N_{j,i}$ is the airborne plant concentration (CN m$^{-3}$) in the $i$th particle size interval of the $j$th sampling month measured by the Andersen sampler, and $F_j$ is the flux density of dry deposition (CN cm$^{-2}$ month$^{-1}$) measured for the $j$th month by the dry deposition sampler. August 2015 data were excluded as air sampling failed.

**3 Results**

**3.1 Particle concentrations in air**

Air and surface deposit samples were collected in Seoul in South Korea. From 18 air sample libraries, 552,074 high-quality ITS2 sequence reads were obtained and mapped onto 97% OTUs (Table S1 in the Supplement). The α-diversity measures of plant assemblages in air samples are listed in Table S2 in the Supplement. Approximately 96% of sequences belonged to the Streptophyta, a plant superdivision that includes terrestrial plants. The remaining 4% of the sequences belonged to the

Chlorophyta, which comprises aquatic organisms such as green microalgae (e.g., Trebouxiophyceae).

The annual mean particle size-integrated concentration of all plant taxa was 133,500 CN m$^{-3}$. The three most dominant classes or clades found in air samples were Pinidae, rosids, and asterids (Fig. 1a), with respective annual mean concentrations of 117,400, 8,600, and 6,400 CN m$^{-3}$ based on the number of ITS2 copies. Taxonomic structures varied with season, albeit not significantly (P-test, ParScore = 9, $p = 0.12$) (Fig. 1b). The highest particle concentrations for Pinidae were

observed in May, whilst for asterids, the highest concentration was observed in September (Fig. 1a).

Genus ranking showed that *Pinus*, *Humulus*, and *Ambrosia* were the three dominant genera found in the air samples (Fig. 2), with respective annual mean concentrations of 116,000, 7,400, and 5,900 CN m$^{-3}$. Species of these genera were confirmed to inhabit the region near the Seoul sampling site. The highest concentrations were observed in May for *Pinus* and in September for *Humulus* and *Ambrosia* (Fig. 2a). These genera contain known human allergenic species (Table 1).



## 3.2 Particle concentrations in air

From 25 deposition sample libraries, 284,703 and 424,795 high-quality ITS2 sequence reads were obtained and mapped onto 97% OTUs for 11 dry deposition and 14 wet deposition libraries, respectively (Table S1 in the Supplement). The α-diversity measures of plant assemblages in deposition samples are listed in Table S2 in the Supplement. In dry deposition samples,
89% and 11% of the sequences belonged to the Streptophyta and Chlorophyta, respectively. In wet deposition samples, 86% and 13% of the sequences belonged to the Streptophyta and Chlorophyta, respectively.

The annual mean flux densities of all plant taxa were 122,000 and 19,000 CN cm$^{-2}$ month$^{-1}$ for dry and wet deposition samples (Fig. 3a, b), comprising 87% and 13% of the total plant particle deposits, respectively. The relative contribution of wet deposition to total deposition appeared to be associated with precipitation levels, except for the peak
contribution preceding the peak precipitation by one month (Fig. 3c). Taxonomic richness in wet deposition increased when precipitation levels were higher, e.g., in June, July, and November (Fig. 3d). By contrast, the taxonomic richness in dry deposition increased when precipitation levels were lower, e.g., in May and September (Fig. 3d). The assemblage structures varied significantly with season (P-test, ParScore = 12, $p < 0.05$), but not with atmospheric deposition mode (P-test, ParScore = 6, $p = 0.17$) (Fig. 3e).

Class ranking showed that Pinidae was the most abundant class in both dry and wet deposition (Fig. 3a, b), with respective annual mean flux densities of 101,000 and 18,000 CN cm$^{-2}$ month$^{-1}$. The second and third most abundant clades were rosids and asterids, respectively, with respective flux densities of 10,900 and 8,100 CN cm$^{-2}$ month$^{-1}$ for dry deposition and 430 and 50 CN cm$^{-2}$ month$^{-1}$ for wet deposition.

Genus ranking showed that *Pinus*, *Humulus*, and *Ambrosia* were the three dominant genera in dry deposition (Fig.
4), with respective flux densities of 100,600, 9,900, and 4,300 CN cm$^{-2}$ month$^{-1}$. In wet deposition, the three dominant genera were *Pinus*, *Juglans*, and *Humulus*, with respective flux densities of 17,700, 170, and 160 CN cm$^{-2}$ month$^{-1}$. Relative dry and wet deposition contributions of selected genera with known allergenic species are shown in Fig. 5. Dry deposition was the predominant mode of atmospheric deposition for allergenic genera: 85.0% for *Pinus*, 98.4% for *Humulus*, 99.3% for *Ambrosia*, 99.6% for *Artemisia*, 92.1% for *Robinia*, and 68.5% for *Quercus* (Fig. 5).

## 3.3 Dry deposition velocities

Dry deposition velocities of selected plant genera were calculated according to Equation 1 (Table 2). The overall velocity of all plant taxa combined was 0.40 cm s$^{-1}$. Little correlation was found between dry deposition velocities and microscopy-based pollen sizes, with a Pearson correlation coefficient of 0.11 (Fig. 6). At the class-level, however, deposition velocities appeared to be taxon-dependent, i.e., 98 cm s$^{-1}$ for Trebouxiophyceae, 2.8 cm s$^{-1}$ for Liliopsida, 0.54 cm s$^{-1}$ for asterids, 0.50
cm s$^{-1}$ for Klebsormidiophyceae, 0.49 cm s$^{-1}$ for Marchantiopsida, 0.44 cm s$^{-1}$ for rosids, 0.39 cm s$^{-1}$ for Pinidae, and 0.027 cm s$^{-1}$ for Bryopsida.



## 4 Discussion

Large quantities of biological particles are emitted into the global atmosphere, with estimates of 0.75 Tg yr$^{-1}$ for bacteria, 31 Tg yr$^{-1}$ for fungi, and 47 Tg yr$^{-1}$ for pollen (Hoose et al., 2010). The emitted particles are involved in global cycling of substances, including the bioprecipitation cycle in which organisms emit airborne particles (or are emitted as airborne

particles) that serve as cloud nuclei and promote precipitation (Morris et al., 2014;Sands et al., 1982). The bioprecipitation cycle can also enhance the environmental conditions for the organisms involved. For example, Woo et al. (2018) reported that fungal basidiospores were deposited predominantly in wet form, while Elbert et al. (2007) indicated that basidiospores were discharged preferentially under humid conditions. This suggests that fungus-mediated bioprecipitation (mycoprecipitation), in which fungi discharge spores that can serve as cloud nuclei and promote precipitation, can create

humid conditions that facilitate spore dispersal. In this study, deposition of plant materials from the atmosphere was examined to determine whether similar mechanisms were present for plants.

Seasonal patterns were observed for plant assemblages in the atmosphere (Fig. 1b) and in deposition (Fig. 3e). The highest concentrations were observed in May for *Pinus* and in September for *Humulus* and *Ambrosia* (Fig. 2a). This correlated with the pollen calendar in Korea (Oh et al., 2012) and suggested that most plant DNA detected in this study was

likely pollen-derived. *Pinus* was the most abundant genus (Fig. 2b), consistent with previous microscopy-based studies in Korea (Jung and Choi, 2013;So et al., 2017). However, DNA-based analysis showed that *Pinus* comprised 87% of the total plant assemblage (Fig. 2b), higher than the contributions estimated by microscopy-based analysis (42–72%) (Jung and Choi, 2013;So et al., 2017). This difference might be due to between-study variabilities in local floral and meteorological characteristics. Another possibility is quantitation biases in DNA metabarcoding, such as biases associated with variation in

the number of ITS copies per pollen grain (Bell et al., 2019). Nonetheless, DNA-based measurements were shown to be reproducible (Fig. S1 in the Supplement), and therefore were suitable for accurate between-sample comparisons.

Modes of pollination differ by plant taxa. Abiotic pollination mechanisms, such as wind pollination (anemophily), are employed by approximately 20% of angiosperms, with biotic pollination such as insect pollination (entomophily) accounting for the remaining 80% (Ackerman, 2000). Some anemophilous angiosperms are arboreal, with examples

including *Juglans*, *Platanus*, *Quercus*, and *Acer* (Molina et al., 1996). *Pinus* is a genus of anemophilous gymnosperms. Most of the genera detected in this study were anemophilous land plants, but entomophilous genera including *Robinia* (Cierjacks et al., 2013) were also detected in small numbers (Table S3 in the Supplement). However, ambophily is known to occur for some angiosperm families (Culley et al., 2002). Some bryophytes (mosses), such as *Streblotrichum*, were also detected (Fig. 2). Bryophytes are known to release spores under desiccated conditions by capsule opening (Gallenmüller et al., 2017).

Dry deposition velocities were calculated according to Equation 1 under the assumption that deposition was materially balanced with the airborne quantity in a well-mixed closed system with a sufficient deposition time. Calculated dry deposition velocities were not meaningfully correlated with microscopy-based pollen sizes (Fig.6), and velocities were



lower than those reported in chamber-based experiments (Table 2). This discrepancy might be attributable to the mass balance assumption used in this study, as it is possible that pollen emitted locally might have dispersed and settled at a remote location outside the system boundary used for our mass balance assumption. The assumptions used for the physical properties of pollen might also have caused discrepancies. Pollen can be desiccated, ruptured, and/or fragmented (Franchi et
al., 2011;Miguel et al., 2006), which can change aerodynamic properties in the air and impact deposition velocities. Irregularly shaped pollen grains, such as spikes of *Ambrosia* and air-filled sacci of *Pinus*, might also confound the relationship between their microscopy-based sizes and aerodynamic properties (Schwendemann et al., 2007;Sabban and van Hout, 2011).

The differences in settling velocities between plant classes or clades might be explained by the differences in pollen
morphology. For example, the larger settling velocity observed for Liliopsida (2.8 cm s$^{-1}$) might be because plants belonging to this class produce pollen with unicolpate structures, while the smaller settling velocities observed for rosids (0.44 cm s$^{-1}$) and asterids (0.54 cm s$^{-1}$) might be because plants belonging to these clades produce tricolpate pollen structures, which result in larger frictional resistance in the atmosphere. The smallest settling velocity observed for Bryopsida (mosses) (0.027 cm s$^{-1}$) might be because the spores they produce (8–40 μm) (Zanatta et al., 2016;Hill et al., 2007) are smaller than the pollen
grains produced by Spermatophyta (10–100 μm) (Hinds, 1999;Jacobson and Morris, 1976).

Contrasting tendencies were observed in the modes of atmospheric deposition between fungi and plants. Unlike fungal particles, which deposited mostly in wet form (86%) (Woo et al., 2018), plant particles deposited predominantly in dry form (87%) (Fig. 3a, b). Moreover, there were no distinct differences in assemblage structures between dry and wet deposition for plants (Fig. 3e), whereas significant differences were observed for fungi (Woo et al., 2018). For example,
Woo et al. (2018) reported that spores from mushroom-forming fungi were highly enriched in precipitation, suggesting that such fungal spores served as nuclei in clouds and/or were discharged preferentially during precipitation. The lack of taxon-dependent tendencies for release of plant particles suggests that the majority of plant species are not specifically dependent on or involved in precipitation. Indeed, most plants and bryophytes (mosses) release pollen or spores by anther or capsule opening under dehydrated conditions (Firon et al., 2012;Gallenmüller et al., 2017). This general xerophytic nature of pollen
dispersal might partially explain why plants were not generally involved in precipitation.

Wet deposition can occur by washout (below-cloud scavenging), which is likely taxon-independent, and rainout (within-cloud scavenging), which is likely taxon-dependent, as observed previously for deposition by ice nucleation-active bacterial and fungal species (Failor et al., 2017;Pouleur et al., 1992). The minimal differences in plant assemblage structures between dry and wet deposition (Fig. 3e) indicated that washout predominated over rainout for wet deposition of
atmospheric plant particles. Large pollen grains might be less likely to reach cloud base altitudes than smaller biological particles such as fungal spores. Cáliz et al. (2018) demonstrated that plant particles were scarce (<10%), and fungal particles were abundant (>75%), in precipitation collected at an altitude of 1,800 m in Spain. Despite this, pollen is not necessarily



insignificant in precipitation as pollen grains can reach cloud base altitudes of 500–2,000 m (Damialis et al., 2017), albeit in attenuated quantities (Noh et al., 2013). Small quantities of large pollen grains might contribute to initiate precipitation since they serve as GCCN that can disproportionately efficiently scavenge smaller droplets in clouds (Johnson, 1982;Möhler et al., 2007).

5       Meteorological conditions such as rainfall are known to be linked to allergic symptoms, for example with so-called thunderstorm asthma in pollinosis patients (D'Amato et al., 2007;D'Amato et al., 2012), but causality remains unclear. D'Amato et al. (2012) suggested that rainfall can have antagonistic effects, removing allergenic pollen grains from the atmosphere but also increasing the abundance of respirable fragments that are released from ruptured pollen grains by osmotic pressure. The present study showed that precipitation was a minor mode of atmospheric deposition of allergenic

pollen grains (Fig. 5). Allergenic pollen might be undispersed and thus depleted from the atmosphere at the time of precipitation since pollen dispersal generally occurs under dry conditions (Firon et al., 2012). However, a small fraction of pollen (~10%) was precipitated (Fig. 5), supporting the proposal that pollen can interact with water droplets in the atmosphere and lead to release of allergenic fragments from moisture-ruptured pollen grains.

      Trebouxiophyceae was detected in the air and deposition samples (Figs. 1 and 3). The most abundant genus was

*Trebouxia* (Fig. 4), which is a desiccation-tolerant aeroterrestrial alga (Candotto Carniel et al., 2015) that is thought to disperse asexual propagules (Ahmadjian, 1988). Previous reports indicate detection in the air (Schlichting, 1969) and in precipitation (Cáliz et al., 2018). Trebouxiophyceae is also found in sea water (Tragin and Vaulot, 2018) and might therefore also be dispersed in the form of sea spray from aqueous ecosystems (Tesson et al., 2016;Mayol et al., 2014). Cáliz et al. (2018) reported enrichment in rainfall at sample sites where aerosols of marine origin were dominant. Another possibility is

that asexual propagules of terrestrial origins might be precipitated by washout and/or rainout as cloud nuclei (Sassen et al., 2003). The present study also showed detection in dry deposits (Fig. 4), indicating the deposition of asexual propagules of terrestrial origins, and/or aqueous cells dispersed as droplets and subsequently desiccated after long-range transport in the atmosphere (Mayol et al., 2017). The high settling velocity (= 99 cm s$^{-1}$) (Table 2) indicated deposition as agglomerates.

## 5 Conclusion

This study showed that dry deposition was the predominant mode of atmospheric deposition of plant particles, including allergenic genera (Fig. 5). This was likely due to the general xerophytic nature of pollen dispersal and rapid settlement of large-size pollen grains. A small fraction (~15%) precipitated via rainout (in-cloud scavenging) and/or washout (below-cloud scavenging). Plant assemblage structures did not differ significantly between dry and wet deposition, indicating that taxon-independent washout predominated over taxon-dependent rainout for wet deposition of atmospheric plant particles. A small

number of plant genera was detected only from wet deposition (Fig. 4), and some genera were detected in wet deposition



with relatively large contributions, e.g., 60% for *Juglans*, and 32% for *Quercus* (Fig. 5). This suggests that these particular genera might be involved in precipitation by serving as nucleation-active species in the atmosphere. Indeed, previous reports showed that a small group of plants discharged pollen grains under rain conditions (rain pollination) (Fan et al., 2012;Hagerup, 1950), suggesting the existence of plant-mediated bioprecipitation. Further interannual monitoring will be required to clarify the deposition tendencies of different plants by controlling for the seasonality of atmospheric plant assemblages observed at our sampling site (Fig. 3e). Additionally, chamber-based experiments are needed to test their nucleation potentials. Finally, we propose that global monitoring be employed to explore for the presence of endemic species that might be specifically involved in plant-mediated bioprecipitation in their regional ecological systems.

**Data availability**

Raw sequence data are available under the project number PRJNA525749 of the NCBI Sequence Read Archive.

**Author contribution**

N.Y. designed the research. K.D. and C.W. performed the research and analyzed data. K.D. and N.Y. wrote the paper. All authors participated in editing the final version of this manuscript

**Competing interests**

None to declare.

**Acknowledgements**

This research was supported by the Basic Science Research Program through the National Research Foundation of Korea (2013R1A1A1004497).

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



**Table 1: Examples of plant genera with known allergenic species.**

| Class or clade | Genus | Common name | Example of allegenic species (ref.) |
|---|---|---|---|
| asterids | *Ambrosia* | Ragweed | *Ambrosia trifida* (D'Amato et al., 2007) |
| | *Artemisia* | Mugwort | *Artemisia vulgaris* (D'Amato et al., 2007) |
| Liliopsida | *Dactylis* | Orchard grass | *Dactylis glomerata* (D'Amato et al., 2007) |
| | *Lolium* | Ryegrass | *Lolium perenne* (Davies, 2014) |
| | *Poa* | Blue grass | *Poa pratensis* (Davies, 2014) |
| | *Triticum* | Wheat | *Triticum aestivum* (Davies, 2014) |
| Pinidae | *Pinus* | Pine | *Pinus radiata* (Gastaminza et al., 2009) |
| rosids | *Acer* | Maple | *Acer rubrum* (White and Bernstein, 2003) |
| | *Betula* | Birch | *Betula papyrifera* (White and Bernstein, 2003) |
| | *Humulus* | Hop | *Humulus japonicus* (Park et al., 1999) |
| | *Juglans* | Walnut | *Juglans nigra* (White and Bernstein, 2003) |
| | *Prunus* | Peach | *Prunus persica* (Pérez-Calderón et al., 2017) |
| | *Quercus* | Oak | *Quercus albus* (White and Bernstein, 2003) |
| | *Robinia* | Locust | *Robinia pseudoacacia* (Kespohl et al., 2006) |
| | *Rosa* | Rose | *Rosa rugosa* (Demir et al., 2002) |
| undefined | *Amaranthus* | Pigweed | *Amaranthus retroflexus* (White and Bernstein, 2003) |
| | *Chenopodium* | Goosefoot | *Chenopodium album* (White and Bernstein, 2003) |
| | *Platanus* | Sycamore | *Platanus occidentalis* (White and Bernstein, 2003) |


**Table 2: Dry deposition velocities and microscopy-based pollen sizes of selected plant genera [a].**

| Class or clade | Genus | Deposition velocity (cm s[-1]) | | Microscopy-based pollen diameter or length × width (ref.) |
|---|---|---|---|---|
| | | This study | Chamber-based studies (ref.) | |
| Asterids | *Ambrosia* | 0.30 | 0.82 cm s[-1] for *Ambrosia trifida* (Durham, 1946b) | 10–25 µm for *Ambrosia artemisiifolia* (Sam and Halbritter, 2016) |
| | *Artemisia* | 2.7 | 1.0 cm s[-1] for *Artemisia annua* (Durham, 1946b) | 10–25 µm for *Artemisia glacialis* (Halbritter and Weis, 2016) |
| Klebsormidiophyceae | *Interfilum* | 3.7 | n.a. | 3.7–7.9 µm for algal cells of *Interfilum* spp. (Karsten et al., 2014) |
| Liliopsida | *Dactylis* | 40 | 2.8 cm s[-1] for *Dactylis glomerata* (Durham, 1946b) | 26–50 µm for *Dactylis glomerata* (Halbritter, 2016c) |
| | *Digitaria* | 7.5 | n.a. | 32–45 µm for *Digitaria exilis* (Damialis and Konstantinou, 2011) |
| | *Lolium* | 6.9 | 4.5 cm s[-1] for *Lolium perenne* (Borrell, 2012) | 26–50 µm for *Lolium perenne* (Halbritter et al., 2015) |
| | *Poa* | 2.9 | 1.5–1.7 cm s[-1] for *Poa pratensis* (Durham, 1946b) | 26–50 µm for *Poa angustifolia* (Diethart, 2016a) |
| | *Setaria* | 0.72 | n.a. | 33 µm for *Setaria viridis* (Douglas et al., 1985) |
| | *Triticum* | 2.4 | n.a. | 51–100 µm for *Triticum aestivum* (Diethart, 2016b) |
| Marchantiopsida | *Marchantia* | 0.38 | n.a. | n.a. |
| Pinidae | *Pinus* | 0.39 | 2.5 cm s[-1] for *Pinus sylvestris* (Durham, 1946b) | 56 × 39 µm for *Pinus koraiensis* (Song et al., 2012) |
| Rosids | *Acer* | 7.7 | n.a. | 26–50 µm for *Acer tataricum* (Halbritter, 2016b) |
| | *Amorpha* | 0.46 | n.a. | 10–25 µm for *Amorpha fruticosa* (Halbritter, 2016d) |
| | *Betula* | 0.19 | 1.7 cm s[-1] for *Betula nigra* (Durham, 1946b) | 10–25 µm for *Betula pendula* (Halbritter and Diethart, 2016a) |
| | *Humulus* | 0.45 | n.a. | 10–25 µm for *Humulus lupulus* (Halbritter, 2016e) |
| | *Juglans* | 0.35 | 2.8 cm s[-1] for *Juglans nigra* (Durham, 1946b) | 26–50 µm for *Juglans regia* (Halbritter and Sam, 2016a) |
| | *Medicago* | 0.006 | n.a. | 26–50 µm for *Medicago falcata* (Halbritter and Svojtka, 2016) |
| | *Prunus* | 0.20 | n.a. | 26–50 µm for *Prunus avium* (Halbritter, 2016a) |
| | *Quercus* | 0.13 | 1.8 cm s[-1] for *Quercus macrocarpa* (Durham, 1946b) | 26–50 µm for *Quercus robur* (Diethart and Bouchal, 2018) |
| | *Robinia* | 0.52 | n.a. | 26–50 µm for *Robinia pseudacacia* (Halbritter and Sam, 2016b) |
| | *Rosa* | 29 | n.a. | 30 × 28 µm for *Rosa rugosa* (Żuraw et al., 2015) |
| Trebouxiophyceae | *Trebouxia* | 99 | n.a. | 4–5 × 3 µm for algal cells of *Trebouxia incrustata* (Peksa and Škaloud, 2008) |
| Undefined | *Amaranthus* | 3.6 | 1.9 cm s[-1] for *Amaranthus palmeri* (Durham, 1946b) | 21–38 µm for *Amaranthus palmeri* (Sosnoskie et al., 2017) |
| | *Chenopodium* | 4.4 | n.a. | 26–50 µm for *Chenopodium album* (Diethart, 2016c) |
| | *Platanus* | 0.053 | 1.0 cm s[-1] for *Platanus occidentalis* (Durham, 1946b) | 26–50 µm for *Platanus hispanica* (Halbritter and Diethart, 2016b) |

a Genera shown in Figs 2 and/or 4 are listed. Genera were excluded if they were undetected in dry deposition or air samples.
n.a., not available in the literature.



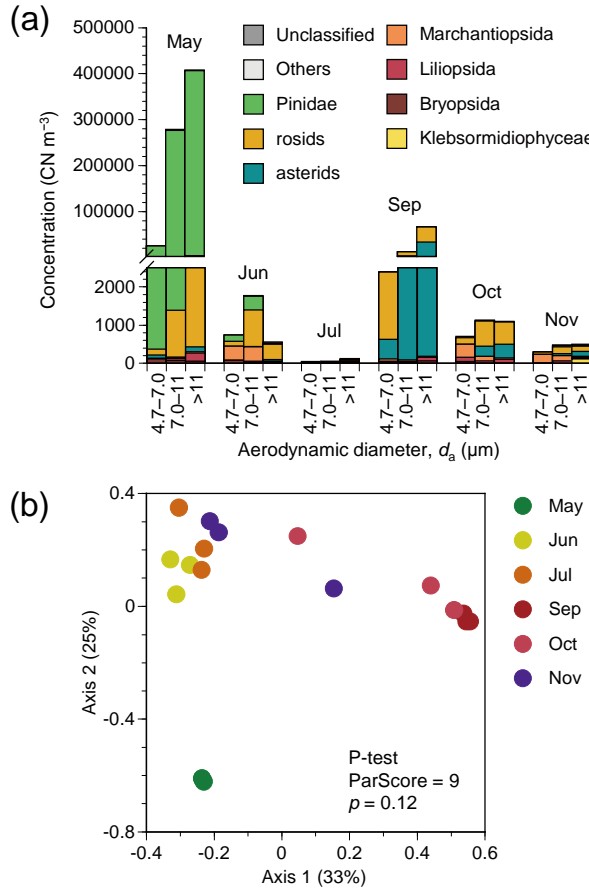

**Figure 1: (a) Particle size-resolved concentrations based on plant classes or clades in terms of copy number (CN) of ITS2 from atmospheric samples from Seoul in South Korea. Monthly results from May to November 2015 are shown, except for August when air sampling failed. (b) Principal coordinate analysis plot for plant assemblage structures based on Bray-Curtis distance.**



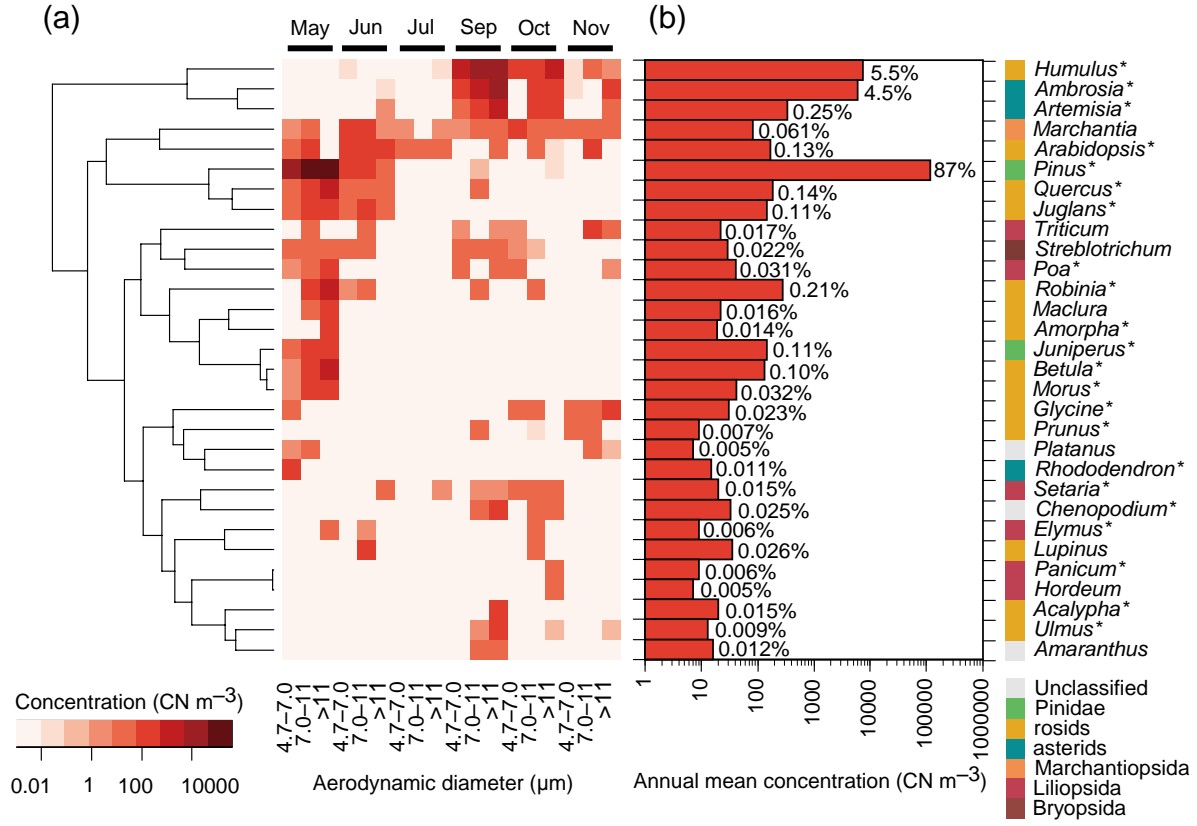

**Figure 2: The 30 most abundant plant genera in terms of ITS2 copy number (CN) in atmospheric samples from Seoul in South Korea.** Asterisks (*) indicate spermatophyte genera that are known to inhabit South Korea and that are listed in the databases of the Korea National Arboretum (2017) and/or the Korea Research Institute of Bioscience and Biotechnology (2016). (a) Monthly particle size-resolved concentrations from May to November 2015 are shown, except for August when air sampling failed. The tree represents the similarities, based on Euclidean distance, of the log-transformed concentrations. (b) Annual mean particle size-integrated concentrations. Percentage values indicate relative contributions.



**Figure 3: Deposition flux densities based on plant classes or clades in terms of copy number (CN) of ITS2 from May to November 2015 in Seoul in South Korea. (a) Dry deposition. (b) Wet deposition. (c) Precipitation and relative contributions of wet deposition to total deposition. (d) Estimated number of 97% OTUs in dry and wet deposition based on the Chao1 estimator. (e) Principal coordinate analysis plot for plant assemblage structures based on the Bray-Curtis distance.**



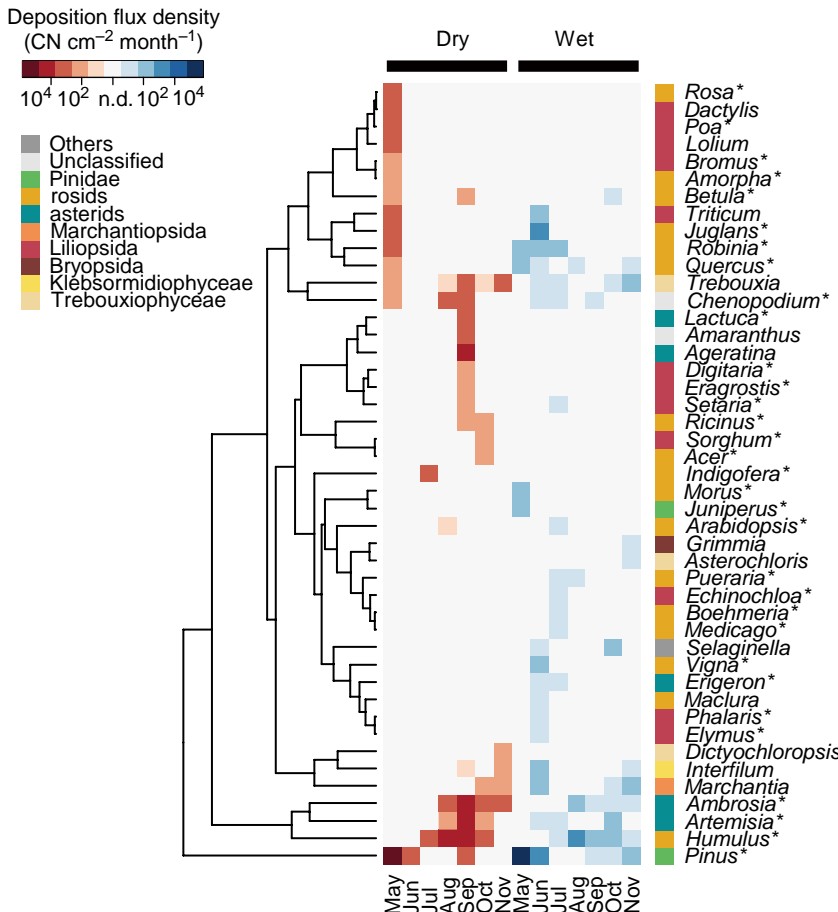

**Figure 4: Deposition flux densities of plant genera in terms of copy number (CN) of ITS2 in Seoul in South Korea. The 30 most abundant plant genera in dry and/or wet deposition are shown. Asterisks (*) indicate the spermatophyte genera that are known to inhabit South Korea and that are listed in the databases of the Korea National Arboretum (2017) and/or the Korea Research Institute of Bioscience and Biotechnology (2016). The tree represents the similarities, based on Euclidean distance, of the log-transformed deposition flux densities.**





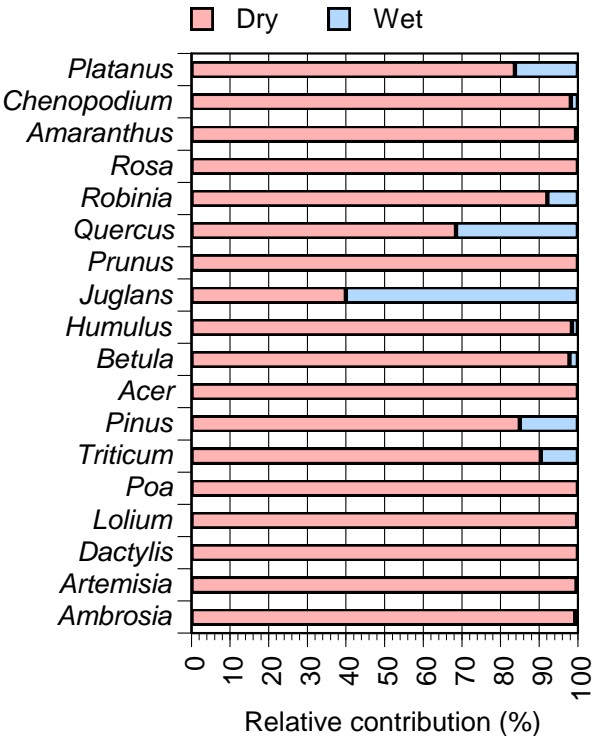

**Figure 5: Relative contributions of dry and wet deposition for the selected plant genera with known allergenic species. Genera in Table 2 are shown, with information regarding allergenic species available in Table 1.**





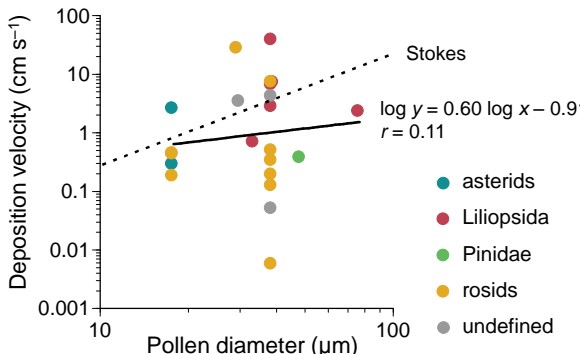

**Figure 6: Dry deposition velocities of the plant genera listed in Table 1. Each point represents a datum for each genus. *Interfilum* and *Trebouxia*, which are algal genera, were excluded. For each genus, a mean value of lower and upper values of reported pollen diameters is used for a representative pollen diameter. In case pollen grains are non-spherical, a mean value of the reported length and width is used. The Stokes terminal gravitational settling velocities are calculated by assuming that pollen grains are spherical and have a standard density of 1.0 g cm$^{-3}$. The velocities are calculated by: $V_{Stk} = \rho_0 d^2 g / 18\eta$ where $\rho_0$ is the standard density (= 1.0 g cm$^{-3}$), $d$ is the pollen diameter, $g$ is the acceleration of gravity (= 980 cm s$^{-2}$), and $\eta$ is the viscosity of air (= $1.8 \times 10^{-5}$ Pa s).**

