# Peer review of "Plant assemblages in atmospheric deposition"

_Atmospheric Chemistry and Physics, 2019_

## Referee Comment (RC1) · Anonymous Referee #1 · 8 Jul 2019

General Comments:

The authors report a wide-ranging study on the two basic types of deposition for aerosol plant particles. They use high-throughput sequencing for identification and additionally qPCR for quantification of the different samples investigated in this study. The adaption of the clustering of OTUs with a similarity of 97% as known for bacteria might for plants be seen critical. References from literature, showing that this is a working method for clustering plant OTUs, are missing. The reached sequencing depth is sufficient as indicated via rarefaction curves and subsampling at 6,142 reads is comprehensible. The methods concerning the qPCR are in good agreement with the standard. Especially the deposition flux calculations give interesting results, but the authors name the weaknesses of the method like variation in the number of ITS gen regions and others (P7 L20). The scientific methods and assumptions are clearly outlined and not redundant. In general, the data of this manuscript are helpfull and this paper might fits to the

scope of ACP, but my personal point of view is that it is more related to the main topic of BGS. The overall writing of the paper appears somehow tedious, mainly because of redundant parts within introduction and discussion, whereas the story could be very attractive with some more focus on the needs of the reader. I want to name two examples comparing the beginning of the introduction and discussion as both start in the same manure, to strengthen my point:

Specific comments:

The authors give the global atmospheric estimates of emitted particles in Tg per year. Noticeable here is that on P1L25 the amount of released plant particles is given with 47-84. In the Discussion P7L2 no range is given but the already mentioned amount of pollen from Hoose et al. 2010 is given with 47 Tg.

P1L25 An estimated 47–84 Tg of plant particles are released into the environment each year (Després et al., 2012;Hoose et al., 2010;Jacobson and Streets, 2009),. . .

P7L2 Large quantities of biological particles are emitted into the global atmosphere, with estimates of 0.75 Tg yr-1 for bacteria, 31 Tg yr-1 for fungi, and 47 Tg yr-1 for pollen (Hoose et al., 2010)

Another example:

P2 L1-5 . . .and/or by serving as ice nuclei (IN) and cloud condensation nuclei (CCN) (Pöschl et al., 2010;Pope, 2010). Finally, atmospheric pollen is involved in globalcycling of substances (Després et al., 2012) by long-range transport and subsequent settlement to the planetary surface (pedosphere) by dry or wet deposition, i.e.,sedimentation or precipitation, respectively.

P7 L2-6 The emitted particles are involved in global cycling of substances, including the bioprecipitation cycle in which organisms emit airborne particles (or are emitted as airborne particles) that serve as cloud nuclei and promote precipitation (Morris et al., 2014;Sands et al., 1982)..

Further question:

I wonder why the results of the Anderson sampler especially for the Pinidae (found within all chambers for all aerodynamic diameter) is not discussed by the authors as it might indicate pollen rapture or the co-emittance of non-pollen particles? Or is this an effect due to the mentioned air-filled sacci of for example Pinus?

Technical corrections:

In Table 1 : asterids and rosids are not capitalized in table 2 they are. Please unify.

Table 2: Please optimize the dimension of the table in a way that no single characters appear as for "Chenopodium".

Figure 1: A method is missing, giving some information how data for the plots were generated. I guess 1A) qPCR and 1B) NGS? This should be added to the caption. Overall seems this figure caption somehow unfinished when compared to all others and scientific standards. One opening sentence would improve this.

---

## Referee Comment (RC2) · Anonymous Referee #2 · 9 Jul 2019

General Comments:

In the paper the authors examine via genetic techniques the deposition flux of plant derived material in Korea. The topic is relevant and interesting, especially considering the potential climatic interactions of primary biogenic aerosols (PBAPs). Any further insight on PBAPs emissions and deposition is for sure a much needed information. The paper is well written and clear, however the reviewer would like some more clarification about the sampling strategy used to compute deposition fluxes and deposition velocities in the paper.

Specific Comments:

Page 3, Lines 5-7: How did the building height compares with the forested area around? Was the sampler located significantly above the treetops? A figure showing the samplers and the sampling location would greatly help

[Figure]

Page 3, Lines 10-11: These lines implies that both deposition and concentration samples were taken monthly. Was the Andersen sampler operated continuously for the month? Were there any issue in saturation of the substrates due to overcollection?

Page 3, Line 12: What are the specifications of such custom-made sampler? The geometry of the collector do impact the deposition process, so how was this custom made sampler validated? In the reviewer's view these are needed information that are lacking also in the referenced Woo et al., 2018 and Han et al., 2016 papers (referred in Woo et al., 2018 regarding the custom made sampler).

Page 3, Line 13: How far were the deposition sampler and the Andersen one? If they were co-located too close to each other, the active air sampling of the Andersen could affect the deposition on the custom made sampler. Again a figure of the sampling setup would greatly help instead of referring to Woo et al., 2018 (in which the figure of the sampler is in the supplementary materials).

Page 4, Lines 11-12: There is a mismatch in the units for flux and concentration. The flux is stated to be reported in CN cm-2 month-1, but the concentration is stated to be measured in CN m-3.

Page 6, Lines 8-10: That seems an extremely anomalous result, which is not further discussed. How do the authors explain that? Was the rain sampler from which precipitation data are taken sufficiently close to the deposition experiments or was it far away enough to justify local differences in rainfall amounts?

Page 6, Lines 27-28: Deposition velocities are computed as the ratio between the deposition sampler and the Andersen one. Given simultaneous measurement of the two it is reasonable to expect that, at least for dry deposition, the mass collected on one sampler strongly correlates with the mass collected on the other one (Mohan, 2016). A "decoupling" between the samplers could also explain some issues in computed deposition velocities, were some kind of mass-comparison tests performed on the samplers?

Page 8, Lines 28-30: The reviewer does not really agree, there's no information in this study to support the actual existence of a taxon-dependent rainout for the sampled pollens, nor to support a prevalence of washout over rainout. The lack of differences between wet and dry deposition samples' structures might also be simply due the -lack- of any taxon-dependence to rainout, rather than the more complicated assumption of washout prevalence over taxon-dependent rainout processes. The reviewer suggests rephrasing.

Page 9, Lines 28-29: Again this is a speculation (see previous comment). The reviewer suggests rephrasing.

Cited References: Mohan S. M. (2016) "An overview of particulate dry deposition: measuring methods,deposition velocity and controlling factors", Int. J. Environ. Sci. Technol., 13:387-402.

---

## Author Comment (AC1) · 2 Aug 2019

**FROM:** Dong, K., Woo, C., Yamamoto, N., authors of acp-2019-487 "Plant assemblages in atmospheric deposition"
**RE:** Response to Reviewer #1
**DATE:** August 2, 2019

The authors thank high-quality comments, especially regarding OTU clustering for plant ITS, by the Reviewer #1. Please find our responses to the reviewers' comments. The page numbers in our responses refer to those of our revised manuscript.

Reviewer #1:

**Comment #1: General Comments:**
**The authors report a wide-ranging study on the two basic types of deposition for aerosol plant particles. They use high-throughput sequencing for identification and additionally qPCR for quantification of the different samples investigated in this study. The adaption of the clustering of OTUs with a similarity of 97% as known for bacteria might for plants be seen critical. References from literature, showing that this is a working method for clustering plant OTUs, are missing. The reached sequencing depth is sufficient as indicated via rarefaction curves and subsampling at 6,142 reads is comprehensible. The methods concerning the qPCR are in good agreement with the standard. Especially the deposition flux calculations give interesting results, but the authors name the weaknesses of the method like variation in the number of ITS gen regions and others (P7 L20). The scientific methods and assumptions are clearly outlined and not redundant. In general, the data of this manuscript are helpful and this paper might fits to the scope of ACP, but my personal point of view is that it is more related to the main topic of BGS. The overall writing of the paper appears somehow tedious, mainly because of redundant parts within introduction and discussion, whereas the story could be very attractive with some more focus on the needs of the reader. I want to name two examples comparing the beginning of the introduction and discussion as both start in the same manure, to strengthen my point:**

Response #1: Regarding our selection of 97% as a threshold for OTU clustering of plant ITS, two previous studies have been cited (Page 4 Line 20). We admit that our selection of 97% as a threshold was operational since there is no consensus threshold available for plant ITS. However, Cornman et al. (2015) reported that most plant species were represented by multiple OTUs of ITS at 97% similarity, suggesting that most plant species are representative based on OTU clustering at 97% similarity. This means that the species-level identifications are likely possible with our method, but we restricted our analyses at the genus and family levels due to possible species-level misidentifications caused by the selection of OTU clustering.

References

Cornman, R. S., Otto, C. R. V., Iwanowicz, D., and Pettis, J. S.: Taxonomic characterization of honey bee (*Apis mellifera*) pollen foraging based on non-overlapping paired-end sequencing of nuclear ribosomal loci, PLoS ONE, 10,

e0145365, https://doi.org/10.1371/journal.pone.0145365, 2015.

Núñez, A., Amo de Paz, G., Ferencova, Z., Rastrojo, A., Guantes, R., García, A. M., Alcamí, A., Gutiérrez-Bustillo, A. M., and Moreno, D. A.: Validation of the Hirst-type spore trap for simultaneous monitoring of prokaryotic and eukaryotic biodiversities in urban air samples by next-generation sequencing, Appl. Environ. Microbiol., 83, e00472-00417, https://doi.org/10.1128/aem.00472-17, 2017.

As the Reviewer #1 pointed out, a limitation of DNA-based methods lies in the difficulty in comparing with traditional microscopy-based methods due to variation in the number of ITS copies per pollen grain. Nonetheless, we confirmed that our measurements were reproducible (Fig. S2 in the Supplement), suggesting that between-sample comparisons are accurate at least within our present DNA-based study.

Regarding redundant parts of our manuscript, please see our responses to the Reviewer #1's second and third comments below.

**Comment #2: Specific comments:**
**The authors give the global atmospheric estimates of emitted particles in Tg per year. Noticeable here is that on P1L25 the amount of released plant particles is given with 47-84. In the Discussion P7L2 no range is given but the already mentioned amount of pollen from Hoose et al. 2010 is given with 47 Tg.**
**P1L25 An estimated 47–84 Tg of plant particles are released into the environment each year (Després et al., 2012;Hoose et al., 2010;Jacobson and Streets, 2009),…**
**P7L2 Large quantities of biological particles are emitted into the global atmosphere, with estimates of 0.75 Tg yr-1 for bacteria, 31 Tg yr-1 for fungi, and 47 Tg yr-1 for pollen (Hoose et al., 2010)**

Response #2: The suggested, redundant sentence has been deleted from the discussion section.

**Comment #3: Another example:**
**P2 L1-5… and/or by serving as ice nuclei (IN) and cloud condensation nuclei (CCN) (Pöschl et al., 2010;Pope, 2010). Finally, atmospheric pollen is involved in global cycling of substances (Després et al., 2012) by long-range transport and subsequent settlement to the planetary surface (pedosphere) by dry or wet deposition, i.e., sedimentation or precipitation, respectively.**
**P7 L2-6 The emitted particles are involved in global cycling of substances, including the bioprecipitation cycle in which organisms emit airborne particles (or are emitted as airborne particles) that serve as cloud nuclei and promote precipitation (Morris et al., 2014;Sands et al., 1982).**

Response #3: The suggested, redundant sentence has been deleted from the discussion section.

**Comment #4: Further question:**
**I wonder why the results of the Anderson sampler especially for the Pinidae (found within all chambers for all aerodynamic diameter) is not discussed by the authors**

**as it might indicate pollen rapture or the co-emittance of non-pollen particles? Or is this an effect due to the mentioned air-filled sacci of for example Pinus?**

Response #4: It is not completely clear to us about what is asked in this question, but we assume that the Reviewer #1 asked about the reasons of why we did not discuss about particle size distribution for each plant taxon, including Pinidae, based on our particle size-resolved measurements by the Andersen sampler. In our previous fungal studies (e.g., Yamamoto et al., 2014; Woo et al., 2018), we computed a representative geometric mean of aerodynamic diameters for each fungal taxon based on their particle size distributions characterized by the Andersen sampler. It was possible for fungi since their particle size distributions were normally (i.e., log-normally) distributed with their peaks situated between $d_a = 2.1–11$ μm for most taxa. Meanwhile, size distributions of plant particles were highly skewed and right-truncated with peaks typically situated at $d_a > 11$ μm (Figure 1a), and it is difficult to accurately compute representative aerodynamic diameters with these highly skewed, right-truncated particle size distributions. This is the reason why we discussed our results based on the literature-derived diameter values of pollen grains rather than based on the experimental particle size distribution data that can be but not accurately characterized by particle size-resolved measurements by the Andersen sampler for large-sized plant particles.

References

Woo, C., An, C., Xu, S., Yi, S.-M., and Yamamoto, N.: Taxonomic diversity of fungi deposited from the atmosphere, ISME J., 12, 2051–2060, https://doi.org/10.1038/s41396-018-0160-7, 2018.
Yamamoto, N., Nazaroff, W. W., and Peccia, J.: Assessing the aerodynamic diameters of taxon-specific fungal bioaerosols by quantitative PCR and next-generation DNA sequencing, J. Aerosol Sci., 78, 1–10, https://doi.org/10.1016/j.jaerosci.2014.08.007, 2014.

**Comment #5: Technical corrections:**
**In Table 1 : asterids and rosids are not capitalized in table 2 they are. Please unify.**

Response #5: Corrected (Page 18).

**Comment #6: Table 2: Please optimize the dimension of the table in a way that no single characters appear as for "Chenopodium".**

Response #6: Corrected (Page 18). Thanks for careful reading.

**Comment #7: Figure 1: A method is missing, giving some information how data for the plots were generated. I guess 1A) qPCR and 1B) NGS? This should be added to the caption. Overall seems this figure caption somehow unfinished when compared to all others and scientific standards. One opening sentence would improve this.**

Response #7: The sentences were added to explain how the data were calculated for each panel. The revised caption reads as follows:

"Figure 1: (a) Particle size-resolved concentrations based on plant classes or clades in terms of copy number (CN) of ITS2 from atmospheric samples from Seoul in South Korea. Monthly results from May to November 2015 are shown, except for August when air sampling failed. The data shown are obtained by multiplication of DNA sequencing-derived relative abundance of each family by a total plant concentration measured by the universal plant-specific qPCR assay. (b) Principal coordinate analysis plot for plant assemblage structures based on Bray-Curtis distance. The data shown are based on DNA sequencing." Page 19

---

## Author Comment (AC2) · 2 Aug 2019

**FROM:** Dong, K., Woo, C., Yamamoto, N., authors of acp-2019-487 "Plant assemblages in atmospheric deposition"
**RE:** Response to Reviewer #2
**DATE:** August 2, 2019

The authors thank high-quality comments, especially regarding our sampling strategy, by the Reviewer #2. Please find our responses to the reviewers' comments. The page numbers in our responses refer to those in our revised manuscript.

**Reviewer #2:**

**Comment #1: General Comments:**
**In the paper the authors examine via genetic techniques the deposition flux of plant derived material in Korea. The topic is relevant and interesting, especially considering the potential climatic interactions of primary biogenic aerosols (PBAPs). Any further insight on PBAPs emissions and deposition is for sure a much needed information. The paper is well written and clear, however the reviewer would like some more clarification about the sampling strategy used to compute deposition fluxes and deposition velocities in the paper.**

Response #1: We appreciate the positive appraisal by the Reviewer #2. Regarding our sampling strategy, please see our responses to the Reviewer #2's comments below.

**Comment #2: Specific Comments:**
**Page 3, Lines 5-7: How did the building height compares with the forested area around? Was the sampler located significantly above the treetops? A figure showing the samplers and the sampling location would greatly help.**

Response #2: It is hard to generalize since the sampling point is situated in a mountainous area, with considerable elevational variation. In general, however, the building height is below upland parts but above lowland parts of the area. The sentences were revised and added to provide the following local topographical information:

"Briefly, air and deposition samples were collected on the roof (approximately 20 m above ground level) of a building at an altitude of 105 m above sea level in a mountainous, forested area of Seoul in South Korea (37°27'55.0"N; 126°57'17.7"E). The highest peak (632 m) at which sparse trees exist was situated in the south-southeast of the sampling site at a horizontal distance of ca. 2.3 km." Page 3 Lines 5-8

The positional information (37°27'55.0"N; 126°57'17.7"E) was provided to check the topographical information using internet-based tools, e.g., Google Maps. We wish that it is found by such internet-based tools rather than by providing a new figure in order to minimize the space of the paper. The information is available, for example, by accessing to the following link:

URL
https://www.google.com/maps/place/37°27'55.0"N+126°57'17.7"E/

**Comment #3: Page 3, Lines 10-11: These lines implies that both deposition and concentration samples were taken monthly. Was the Andersen sampler operated continuously for the month? Were there any issue in saturation of the substrates due to overcollection?**

Response #3: The Reviewer #2 is correct. Each sampling continued for a period of 1 month. To prevent from particle overloading, the substrate was rotated once every week for particles to be collected as evenly as possible on the substrate. To clarify, the following sentence has been added:

"The substrate placed onto each stage of the Andersen sampler was rotated once every week to prevent from particle overloading at the same spot under each impactor nozzle." Page 3 Lines 19-21

**Comment #4: Page 3, Line 12: What are the specifications of such custom-made sampler? The geometry of the collector do impact the deposition process, so how was this custom made sampler validated? In the reviewer's view these are needed information that are lacking also in the referenced Woo et al., 2018 and Han et al., 2016 papers (referred in Woo et al., 2018 regarding the custom made sampler).**

Response #4: Photos of the samplers are available in Supplementary Fig. S1 in Woo et al. (2018). The configuration of the dry deposition sampler is identical to that reported by Yi et al. (1996), while the configuration of the wet deposition sampler is similar to that reported by Landis and Keeler (1997). The following sentence has been added for clarification.

"The configuration of the dry deposition sampler is identical to that reported by Yi et al. (1997), while the configuration of the wet deposition sampler is similar to that reported by Landis and Keeler (1997)." Page 3 Lines 17-19

References
Landis, M. S., and Keeler, G. J.: Critical evaluation of a modified automatic wet-only precipitation collector for mercury and trace element determinations, Environ. Sci. Technol., 31, 2610–2615, https://doi.org/10.1021/es9700055, 1997.
Woo, C., An, C., Xu, S., Yi, S.-M., and Yamamoto, N.: Taxonomic diversity of fungi deposited from the atmosphere, ISME J., 12, 2051–2060, https://doi.org/10.1038/s41396-018-0160-7, 2018.
Yi, S.-M., Holsen, T. M., and Noll, K. E.: Comparison of dry deposition predicted from models and measured with a water surface sampler, Environ. Sci. Technol., 31, 272–278, https://doi.org/10.1021/es960410g, 1997.

**Comment #5: Page 3, Line 13: How far were the deposition sampler and the Andersen one? If they were co-located too close to each other, the active air sampling of the Andersen could affect the deposition on the custom made sampler. Again a figure of the sampling setup would greatly help instead of referring to Woo et al., 2018 (in which the figure of the sampler is in the supplementary materials).**

Response #5: The deposition and Andersen samplers were placed distant enough to avoid the interference, with approximate horizontal distance of 2.5 m and vertical distance of 2.3 m (Fig. AC2-1 below, which is also included as Fig. S1 in the revised Supplement). The deposition sampler was placed on a wooden raised floor with approximately 2.3 m height from the rooftop, while the Andersen sampler was placed on the rooftop under the raised floor to protect from precipitation, with an additional rain shield.

[Figure]

**Figure AC2-1:** Schematic diagram showing the sampling setup.

In our revised manuscript, the following sentence has been added to clarify our sampling setup:

"The deposition and Andersen samplers were placed distant enough to avoid the interference, with the approximate horizontal distance of 2.5 m and vertical distance of 2.3 m (Fig. S1 in the Supplement)." Page 3 Lines 16-17

**Comment #6: Page 4, Lines 11-12: There is a mismatch in the units for flux and concentration. The flux is stated to be reported in CN cm$^{-2}$ month$^{-1}$, but the concentration is stated to be measured in CN m$^{-3}$.**

Response #6: The dimension of flux density ($F$) is given by [quantity][area]$^{-1}$[time]$^{-1}$. ($ML^{-2}T^{-1}$), while the dimension of concentration ($N$) is given by [quantity][volume]$^{-1}$ ($ML^{-3}$). The dimension of velocity ($V$) is given by [length]$^{-1}$[time]$^{-1}$ ($LT^{-1}$). We believe that the physical dimensions in our manuscript are correctly given to provide a following relationship of:

$$V = \frac{F}{N} \quad \left( \because \frac{\text{L}}{\text{T}} = \frac{\text{M}/\text{L}^2\text{T}^1}{\text{M}/\text{L}^3} \right)$$

**Comment #7: Page 6, Lines 8-10: That seems an extremely anomalous result, which is not further discussed. How do the authors explain that? Was the rain sampler from which precipitation data are taken sufficiently close to the deposition experiments or was it far away enough to justify local differences in rainfall amounts?**

Response #7: The dry and wet samplers were deployed closely enough with the approximate distances of 25–55 cm (please see Supplementary Fig. S1 in Woo et al. (2018)). Therefore, we believe that it was not due to artifacts associated with the distance between the dry and wet deposition samplers.

We do not know why the peak contribution of wet deposition preceded the peak precipitation by one month (i.e., from July to June) (Fig. 3c), which was anomalous as the Reviewer #2 pointed out. We expect, however, that it was in part attributable to the uncertainty of taking monthly averages for the analyses. Within each month, there were both rainy and non-rainy days. It is possible that some species were released, because of its seasonality, more preferentially during a less rainy month of June than during a rainier month of July even though these species were released more preferentially in rainy days. For instance, we found that *Quercus* and *Juglans* were released in a less rainy month of June, but not in a rainier month of July (Fig. 2a), even though these genera were found abundantly in wet deposition (Fig. 5), indicating that these genera were released preferentially in rainy days of a less rainy month of June. To explain such a possibility, the following paragraph has been added:

"It should be noted, however, that several genera was detected exclusively from wet deposition (Fig. 4), and some allergenic genera were detected abundantly from wet deposition, e.g., 60% for *Juglans*, and 32% for *Quercus* (Fig. 5), indicating that these genera might be specifically involved in precipitation. Additionally, we observed that *Quercus* and *Juglans* were released in a less rainy month of June than in a rainier month of July (Fig. 2a), even though they were detected abundantly in precipitation (Fig. 5), indicating that these genera might be released preferentially in rainy days of a less rainy month of June. We expect that the taxon dependency of seasonal pollen dispersals in conjunction with the taxon dependency of rainfall involvement might partially explain our anomalous observation where the peak contribution of wet deposition preceded the peak precipitation by one month (i.e., from July to June) (Fig. 3c)." Page 9 Lines 7-14

References
Woo, C., An, C., Xu, S., Yi, S.-M., and Yamamoto, N.: Taxonomic diversity of fungi deposited from the atmosphere, ISME J., 12, 2051–2060, https://doi.org/10.1038/s41396-018-0160-7, 2018.

**Comment #8: Page 6, Lines 27-28: Deposition velocities are computed as the ratio between the deposition sampler and the Andersen one. Given simultaneous measurement of the two it is reasonable to expect that, at least for dry deposition,**

**the mass collected on one sampler strongly correlates with the mass collected on the other one (Mohan, 2016). A "decoupling" between the samplers could also explain some issues in computed deposition velocities, were some kind of mass-comparison tests performed on the samplers?**
**Cited References: Mohan S. M. (2016) "An overview of particulate dry deposition: measuring methods, deposition velocity and controlling factors", Int. J. Environ. Sci. Technol., 13:387-402.**

Response #8: The suggested analysis was made, with a good correlation (r=0.91) observed between the dry deposition and Andersen samplers (Fig. AC2-2 below).

[Figure]

**Figure AC2-2:** Relationships between flux densities and concentrations of total plants measured by the dry deposition and Andersen samplers, respectively. (a) Scatter plot. (b) Time-series plot.

The result shows the largest between-method variability observed in July (Fig. AC2-2b above). We checked the taxon-specific results, but could not find any systematic tendencies. It is likely because of the intrinsic measurement uncertainty since the similar variabilities were observed for the duplicate measurements of deposition flux densities in low quantity regions (Fig. S2b and c in the Supplement).

In our revised manuscript, Mohan (2016) has been cited.

"The annual dry deposition velocity ($V_d$) was estimated for each plant taxon according to the following equation (Mariraj Mohan, 2016):" Page 5 Lines 18-20

**Comment #9: Page 8, Lines 28-30: The reviewer does not really agree, there's no information in this study to support the actual existence of a taxon-dependent rainout for the sampled pollens, nor to support a prevalence of washout over rainout. The lack of differences between wet and dry deposition samples' structures might also be simply due the lack of any taxon-dependence to rainout,**

**rather than the more complicated assumption of washout prevalence over taxon-dependent rainout processes. The reviewer suggests rephrasing.**

Response #9: We agree that this is our speculation. The sentence has been revised to clarify that it is just a possibility.

"The minimal differences in plant assemblage structures between dry and wet deposition (Fig. 3e) indicated a possibility that washout, which is possibly taxon-independent, predominated over rainout, which is possibly taxon-dependent, for wet deposition of atmospheric plant particles although it is also possible that there is no taxon dependency in rainout." Page 9 Lines 17-20

**Comment #10: Page 9, Lines 28-29: Again this is a speculation (see previous comment). The reviewer suggests rephrasing.**

Response #10: We agree. It is just a possibility. The sentence has been revised as follows:

"Plant assemblage structures did not differ significantly between dry and wet deposition, indicating a possibility that washout, which is possibly taxon-independent, predominated over rainout, which is possibly taxon-dependent, for wet deposition of atmospheric plant particles." Page 10 Lines 18-20

For the Reviewer #2's 9[th] and 10[th] comments above, we replaced the words "likely" with "possibly" since we do not know the likeliness (although we do know it is possible because the taxon dependency was observed at least for fungi). The sentence in the abstract section was also revised accordingly.

"Plant assemblage structures did not differ significantly between dry and wet deposition, indicating a possibility that washout, which is possibly taxon-independent, predominated rainout, which is possibly taxon-dependent, for wet deposition of atmospheric plant particles." Page 1 Lines 16-18

---

## Author Response (AR2)

**FROM:** Dong, K., Woo, C., Yamamoto, N., authors of acp-2019-487 "Plant assemblages in atmospheric deposition"

**RE:** Response to Editor

**DATE:** August 29, 2019

The authors thank constructive suggestions provided by the Editor. Please find our responses to the Editor's comments. The page numbers in our responses refer to those of our revised manuscript.

**Comment #1:**
**Add a sentence summarizing your response to Response #1 from Reviewer #1, e.g. stating that the 97% threshold was operational in nature and that your usage of this value is already more conservative than what Cornman et al. suggests may be required.**

Response #1: The following sentences have been added in our revised manuscript:

"Our usage of 97% as a threshold was operational since there is no consensus threshold available for plant ITS although Cornman et al. (2015) reported that most plant species were represented by multiple OTUs of ITS at 97% similarity, suggesting that most plant species are identifiable based on OTU clustering at 97% similarity. Nonetheless, we restricted our analyses only down to the genus level to prevent from possible species-level misidentifications." Page 4 Lines 22-26

**Comment #2:**
**Related to Comment #2 from Reviewer #2: The way I read this question, the reviewer is not asking so much about the sampler height in relation to the mountain, but also in relation to the height of the trees as well. Also, I clicked on the link that you included in the response document, and now I can easily see that the measurement site was in what I would consider to be at least semi-urban in nature. I think describing it only as "a mountainous, forested area of Seoul" does not immediately describe what must surely be heavy influence from the populated areas. Please add some more explanation here, i.e. that the site has influence from airflow from the large populated area/city as well as influence from the non-populated and forested regions in the mountains area within the metropolitan area. Is there also a citation that you can add that discusses characterization of the site and aerosol types that can be sampled there? The reviewer suggested making a simple schematic diagram of the relative topography of the sampling site in order to show the height of the trees and buildings related to where the sampler was placed. Given the complexity of the types of land cover types in the immediate area, it could indeed be useful to the reader. If you decide to make the additional figure, you can place in the supplement and only refer to it briefly.**

Response #2: The authors agree with the Editor in regard to how to describe our sampling site. In the revised manuscript, the following explanation has been provided:

"Briefly, air and deposition samples were collected on the roof (approximately 20 m

above ground level) of a building at an altitude of 105 m above sea level in a semi-urban mountainous forested area in the outskirts of a megacity of Seoul in South Korea (37°27'55.0"N; 126°57'17.7"E). The highest peak (632 m) at which sparse trees exist was situated in the south-southeast of the sampling site at a horizontal distance of ca. 2.3 km (Fig. S1 in the Supplement). In addition to biological particles released from the local forested area, the site has influence of abiotic pollutants released from large populated areas in Seoul (Lee, 2014)." Page 3 Lines 5-10

Now, we agree with the Reviewer #2 regarding the necessity of inclusion of a schematic diagram of the relative topography of our sampling site. We have added the supplementary figure showing local topographical information of our sampling site (see below and Fig. S1 in the Supplement).

[Figure]

Figure S1: Topographical information of our sampling site. (a) Topographical map of the Korean peninsula. (b) Topographical map of the local sampling area.

**Comment #3:**
**I specifically asked Reviewer #2 about her/his opinions of the updates to the manuscript. S/he responded with a further comment on Response #6:**
**The dimensional analysis made by the author is correct, however the reviewer still finds a discrepacy when considering units rather than dimensions. The authors, in fact, specify that Vd (the units of which are not stated in the paragraph, but it is expressed in cm s$^{-1}$ in paragraph 3.3) is a ratio between a flux in (CN cm$^{-2}$ month$^{-1}$, page 5 line 22) and a concentration (in CN m$^{-3}$, page 5 line 23). By substituting these units in eq. 1 we will have that:**

**cm s$^{-1}$ = (CN cm$^{-2}$ month$^{-1}$) * (m$^3$ CN$^{-1}$)**

**The reviewer fails to understand how it is possible to obtain a Vd in cm s$^{-1}$ on the left side when the units for time on the right side are in months and the length units are mismatched in the two terms on the right side (cm vs. m). While the reviewers agree that both s and months are dimensionally consistent (as being both times, T) and cm and m are consistent by being both lengths (L), they're far from the same! So either a scaling has been applied to both fluxes (in order to convert them from CN cm$^{-2}$ month$^{-1}$ to CN cm$^{-2}$ s$^{-1}$) and concentrations (in order to convert them to CN m$^{-3}$ to CN cm$^{-3}$), either Vd cannot be expressed in terms of cm s$^{-1}$. If these scalings have been applied, then please explicitate that, otherwise please correct the magnitudes of terms in order for them to be in consistent units and not only in consistent dimensions.**

Response #3: The authors thank the clarification made by the Editor. Now, we understood what was concerned by the Reviewer #2. Yes, we did scalings from m to cm, and from month to second in our calculations. The following clarification has been made in our revised manuscript.

[revised manuscript text omitted]